# The effects of tryptophan loading on Attention Deficit Hyperactivity Disorder in adults: A remote double blind randomised controlled trial

**Larisa M. Dinu[1], Samriddhi N. Singh[1], Neo S. Baker[1], Alexandra L. Georgescu[1], Paul G. Overton[2], Eleanor J. Dommett[1]***

1 Department of Psychology, Institute of Psychiatry, Psychology & Neuroscience, King's College London, London, United Kingdom, 2 Department of Psychology, The University of Sheffield, Cathedral Court, Sheffield, United Kingdom

* Eleanor.dommett@kcl.ac.uk

## Abstract

### Background

Despite the impact and prevalence of Attention Deficit Hyperactivity Disorder (ADHD), current treatment options remain limited and there is a drive for alternative approaches, including those building on evidence of a role for tryptophan (TRP) and serotonin (5-HT). This study aimed to evaluate the effect of acute TRP loading on attention and impulsivity in adults with ADHD.

### Trial design and methods

We conducted a remote double blind randomised controlled trial (RCT) using TRP loading to examine the effects of a balanced amino acid load in comparison to low and high TRP loading in individuals with ADHD (medicated, N = 48, and unmedicated, N = 46) and controls (N = 50). Participants were randomised into one of three TRP treatment groups using stratified randomisation considering participant group and gender using a 1:1:1 ratio. Baseline testing of attention and impulsivity using the Test of Variables of Attention Task, Delay Discounting Task, and Iowa Gambling Task was followed by consumption of a protein drink (BAL, LOW, or HIGH TRP) before repeated testing.

### Results and conclusions

No effects of TRP were observed for any of the measures. In the present study, TRP loading did not impact on any measure of attention or impulsivity in those with ADHD or Controls. The findings need to be confirmed in another trial with a larger number of patients that also considers additional measures of dietary protein, plasma TRP and aggression. (Registration ID ISRCTN15119603).

**Data availability statement:** All data files are available from the Open Science Framework database (DOI.10.17605/OSF.IO/3PY72 and url: https://osf.io/3py72/).

**Funding:** The authors received no specific funding for this work.

**Competing interests:** The authors have declared that no competing interests exist.

## Introduction

Attention Deficit Hyperactivity Disorder (ADHD) is the most common neurodevelopmental condition characterized by heightened inattention, impulsivity and hyperactivity [1]. It affects around 5% of children [2, 3] and 3% of adults worldwide [4]. The impact of ADHD is significant—for example, it is associated with poorer academic outcomes, social difficulties, and lower occupational status [5]. Despite the prevalence and impact of ADHD, treatments remain limited. Psychostimulants are the most effective treatment, reducing symptoms in 80% of patients [6–8]. Although they are thought to primarily act on dopaminergic and noradrenergic systems, evidence also suggests they impact serotonin (5-HT), increasing levels of the transmitter [9, 10]. However, they must be taken continuously to be effective [11] and are associated with various side effects of differing severity, ranging from mild insomnia to tachycardia [12]. Moreover, some patients experience only a 30% symptom reduction, demonstrating treatment needs are not always fully met for all individuals with stimulant medication [13]. Non-stimulants, which primarily impact on the noradrenergic system, offer an alternative medication, but with a lower response rate [14] and a different range of side effects [12].

The regular required use, possible side effects, and residual symptoms often present with current drug treatments that act primarily on the dopaminergic and noradrenergic systems, and the gaps in our knowledge of the aetiology of ADHD, introduces the possibility that other neurotransmitters might be integral and be a suitable target for treatment. One candidate is 5-HT, which may also partially underlie the benefits seen with psychostimulants as indicated above [9]. Furthermore, this transmitter has been consistently linked to ADHD symptoms. For example, studies indicate that 5-HT changes can impact impulsivity [15, 16], regulation of attention [17, 18] and the default mode network (DMN), which is hypothesized to be altered in ADHD [19]. There is also mounting evidence for altered serotonergic genes in the condition [15, 20]. From a treatment perspective, the tricyclic anti-depressant, desipramine, has been shown to be effective in treating symptoms of ADHD [21], and whilst these effects could be due to actions on noradrenaline, histamine or acetylcholine rather than 5-HT, studies with more selective drugs indicate a role for serotonin. For example, a small number of studies examining serotonin-noradrenaline reuptake inhibitors indicate these drugs are effective in ADHD [22, 23] and selective serotoninergic drugs (SSRIs) have been shown to reduce hyperactivity and inattention in individuals with ADHD and comorbid depression [24]. SSRIs are also considered useful in managing the irritability and aggression that can occur with ADHD [25]. Acute administration studies with the SSRI fluoxetine have demonstrated that, in participants with ADHD, the drug modulates the DMN [26], upregulates activity in areas associated with impulsivity, normalises some elements of impulsive behaviour [27], and normalises prefrontal dysfunction [28]. In all these studies effects were reported after a single dose of the SSRI suggesting that even acute uplifts in 5-HT synaptic levels can be beneficial.

Given the involvement of 5-HT in ADHD, including some data suggesting 5-HT drugs may be effective in treating the condition, it is unsurprising that this neurotransmitter has been the focus of other treatment approaches, including dietary interventions. Whilst dietary modifications are not a recommended treatment for ADHD [29], concerns about the side effects, safety and long-term use of psychostimulant medications drive individuals and their families to explore these approaches [30]. Serotonin synthesis and release is dependent on concentration of the precursor molecule, tryptophan (TRP), an essential amino derived from our diet [31]. TRP can be found in high protein foods and is readily available as a powder additive meaning that modulating TRP levels could be relatively straightforward. In support of this, there is evidence that administration of exogenous TRP results in a rise in brain TRP and 5-HT synthesis within the brain [32], which in turn has been shown to result in increased

release of 5-HT in various areas of the brain [33]. Research to date has indicated that lower levels of TRP may be associated with ADHD, at least in adults [34–36] and the gene responsible for encoding tryptophan hydroxylase 2 (TPH2), critical for synthesis of central serotonin, has been linked to ADHD symptoms [37, 38] as well as performance on standardised laboratory tests of attention [39]. Moreover, tryptophan transport mechanisms have also been found to be altered in ADHD, which could result in lower 5-HT levels [40]. Within healthy populations, TRP modulation has also been shown to impact on measures of impulsivity [41, 42].

With the mounting evidence in support of a role for TRP and 5-HT in ADHD, several studies have used dietary modulation of TRP within ADHD populations. A recent systematic review identified 14 studies taking this approach, all of which used acute TRP depletion [43]. This review noted that most studies did not actually measure the core symptoms of inattention, impulsivity, and hyperactivity, but focused on aggressive behaviour, predominantly in children and with male-dominated samples. The results regarding aggressive behaviour, however, were promising, showing that lowering TRP acutely increased aggression, allowing the inference that raising TRP levels may be beneficial. However, reviews focusing on acute TRP depletion [43] and dietary supplementation more generally [44] noted the need for further research into TRP and specifically the need for larger studies, consideration of core symptoms, participants of both genders, and greater representation of adults with ADHD. It was also suggested that modulation approaches other than depletion should be considered, given that TRP levels may be lower in ADHD and as such the effects of further depletion may be confounded by floor effects. Research indicates that lowering TRP will result in lower central 5-HT, and raising TRP will increase 5-HT [45]. Therefore, if those with ADHD have lower levels of 5-HT as has been found [34–36], TRP loading could address this. In line with these suggestions, the current study aimed to investigate, for the first time, the effects of TRP modulation, using a loading paradigm, on attention and impulsivity in adults with ADHD. Given that previous work has indicated effects of acute TRP depletion on ADHD [43] and single doses of SSRI are effective in altering ADHD-related behaviours [26–28], we opted to explore acute loading. We hypothesized that acute TRP loading would reduce inattention and impulsivity in a dose dependent manner in those with ADHD, and therefore may be a suitable treatment for ADHD.

## Methods

### Trial design

This study was a double blind randomised controlled trial with pre and post tryptophan loading measures in participants with a diagnosis of ADHD and those without. We utilised three tryptophan conditions: balanced, low loading and high loading, and distinguished between participants with ADHD who were taking medication (ADHD-M) and those who were not (ADHD-UM). This created a 3 (TRP: BAL, LOW, HIGH) x 3 (Group: Control, ADHD-M, ADHD-UM) by 2 (Time: pre, post) design. Participants without ADHD were included to act as a reference group and establish whether the same effects were found in those without the condition. The study was registered on the ISRCTN registry for clinical trials (ID ISRCTN15119603). There were no changes to the protocol provided although we did re-run power calculations as detailed below (https://doi.org/10.1186/ISRCTN15119603). Reporting aligns with CONSORT guidelines [46].

### Participant recruitment

Participants with and without ADHD were recruited from the community through posters across university campuses, social media adverts, institutional recruitment emails, and

advertisements on appropriate support groups' websites and newsletters (e.g., UK Adult ADHD Network). All advertisements contained a link to the study's information sheet and consent form where interested individuals could provide consent and complete an online screening survey. Recruited participants completed the study in a live online experiment with a researcher. The remote but synchronous approach was required as the start of the study coincided with the COVID-19 pandemic when attendance at the laboratory was not permitted. The entire experiment took around two hours to complete. Participants received a £20 shopping voucher.

No prior research was available measuring attention or impulsivity using a TRP loading paradigm. However, two studies had examined attention in those with ADHD using a TRP depletion paradigm [47, 48] and one using a similar paradigm to the current study reported a large effect size [48]. No studies had examined impulsivity before and after TRP modulation in ADHD except one which primarily focused on hostility, meaning it was not possible to use the data for power calculations [49]. Although we initially assumed a small-to-medium effect ($f = 0.20$), based on effects across several constructs [43], we later revised this using the work by Mette et al. [48], which was closest to our paradigm, to assume a medium effect size ($f = 0.25$). Using G*Power, we calculated the required sample size with an effect size of $f = 0.25$, α (type I error level) = 0.05 and β (type II error level) for a repeated measures ANOVA, powered to detect interaction effects aligning with our hypothesis (i.e., to examine whether TRP modulation had dose dependent effects). Using these parameters, 108 participants would be required in total, which amounts to 12 per group.

## Ethics

This research was approved in advance by the Institutional Ethics Committee (HR-19/20-17983). All participants had access to the information sheet and an online consent form, which they were able to work through at their own pace. Written consent was provided electronically for participation. Recruitment took place between the 22nd November 2020 and the 21st June 2023.

## Eligibility criteria for participants

**Inclusion criteria.** All participants were aged 18–35 years old. The lower age limit was selected given the target population was adults and the upper age-limit was set to avoid age-related cognitive decline which has been found to onset as early as 35 years in some cases [50]. Control participants had no diagnosis of ADHD and scored <14 on the Adult ADHD Self-Report Scale screener items (ASRS-A, items 1–6) [51, 52]. Participants with ADHD were required to have an existing diagnosis of ADHD and score ≥14 on the ASRS-A. Participants with ADHD were included if they were either unmedicated (ADHD-UM), and had been so for at least three months, or were receiving medication for ADHD (ADHD-M), and had been stable on this medication for at least 1 month with at least 70% adherence, as assessed by the screening survey using a previously developed adherence scale [53]. In addition to these criteria, participants had to abstain from consuming caffeinated drinks or alcohol on the day of testing, avoid having protein-rich meals the night before or on the day of testing.

**Exclusion criteria.** Participants were excluded if they had a diagnosis of any nutritional, psychiatric, or neurological illness or a learning difference (e.g., dyslexia). Other exclusion criteria were related to the required consumption of a whey protein drink and were gluten or lactose intolerance or dietary requirements of vegan, halal or kosher as the protein was not suitable for these diets. Individuals were also excluded if pregnant or breastfeeding or a

smoker. Additionally, they were excluded if they were following a particularly restrictive diet (e.g., Keto) or taking medication known to impact on the serotonergic system.

## Procedure

**Online screening and scheduling.** The online screening survey included items to assess inclusion criteria, including age, dietary requirements and intolerances, ADHD symptomology (ASRS), medication use and adherence, and the presence of conditions other than ADHD. In addition, demographic data was collected (gender, ethnicity, handedness, years in post-compulsory education). Note that we opted to ask participants about gender rather than sex in this study and did not collect data on sex separately in order to keep the screening survey short. Results from screening surveys were reviewed within 5 days of completion and all participants emailed. Eligible participants were contacted by email and asked to provide a UK postal address for the protein drink and identify a suitable time for the testing session which would allow them to remain in the online room for around 2 hours with their camera on and in a quiet, undisturbed space. Twenty-four hours prior to this chosen session, participants were sent a reminder of the session and reminded of the guidance to avoid protein-rich meals, caffeine, and alcohol prior to the test session.

**Randomisation.** Using a web-based random number generator, randomisation was stratified by group (Control, ADHD-M, ADHD-UM) and gender (Male, Female) with a 1:1:1 ratio between allocation to the different TRP conditions. Randomisation was conducted by the lead investigator, who enrolled participants to the study, and concealed allocation from those conducting the testing until all testing was complete. Prior experience suggests high drop-out for this participant population between expressing an interest in the study and completing testing and as such we created randomisation tables for up to 20 participants per stratified group i.e., above the required number per group. Participants and researchers conducting testing were blind the tryptophan allocation until all data was collected as packages were sent out without labelling of contents and packages contained identical appearing white protein powder and were the same flavour.

**Testing session and intervention.** On arrival in the online meeting room (Microsoft Teams), participants were reminded of the study information and consent was re-confirmed verbally. They also confirmed that they had not consumed protein-rich meals, caffeine, or alcohol the day of the session or the previous night. Participants were advised to open the protein powder that had been posted to them and mix the powder with 400 ml of water. They were advised in advance of the need to be able to measure out 400 ml of water accurately and did so on camera with the researcher. This was then placed in the fridge until they were instructed to drink it. All protein powders contained a base of vanilla flavoured whey protein (Bulk Powders, UK) and we adopted the loading paradigm utilised previously [54]. The balanced condition (BAL) contained only 40g of 100% whey powder which provided balanced amino acid availability and contained 0.566g of TRP. The low loading condition had added TRP resulting in 1.43g of TRP and the high loading condition contained 5.24g of TRP. As would be expected for a commercially available drink, no adverse effects or harms were reported by participants consuming the drink. The participants were then required to undertake cognitive testing to measure core symptoms of ADHD at baseline (see 'Measures') i.e., before consuming the protein drink. After baseline testing, participants were asked to drink the protein drink within 10 minutes and then wait a further 1 hour before completing the tests again. This interval was chosen as previous research using this paradigm has indicated plasma TRP levels peak at 1 hour after consumption [54] and other, similar work, has also found substantial increases in plasma TRP within one hour [55, 56]. Furthermore,

work in animals has demonstrated a good correlation between plasma and brain levels of TRP following TRP injections [57, 58] or oral administration [59]. All of this was completed in the online meeting room under the supervision of the researcher. After 1 hour had lapsed, cognitive testing was repeated.

**Measures.** The primary outcomes in this trial provided measures of inattention and impulsivity using three established cognitive tests delivered via the online experimental platform, Gorilla.sc [60], which allows standardised instructions to be presented to all participants.

Firstly, we used the Test of Variables of Attention (TOVA) task, a type of Continuous Performance Task (CPT) which has been shown to differentiate those with ADHD from controls [61–63]. It aligns with the current understanding of attention [64, 65] and has been previously used in a study looking at TRP genotype and ADHD symptoms [39]. Furthermore, performance on this task has been found to be impacted by TRP depletion, including in those with ADHD, indicative of performance being dependent on serotonin synthesis [48, 66]. Participants must respond to target ('Go') stimuli and inhibit responses to non-target ('No-Go') stimuli. Stimuli are presented for 100ms, with an inter-stimulus-interval of 2 seconds. There are two phases of the task: i) Phase 1 –Go signals are presented on 22.5% of trials and are therefore infrequent, ii) Phase 2 –Go signals are presented on 77.5% of trials and are therefore frequent. These phases provide information about inattention and motor impulsivity, respectively [64, 67]. Omission errors and hit reaction times from Phase 1 provided measures of attention. d'prime was also calculated for this phase as a measure of response sensitivity related to attention. In Phase 2, commission errors provided information about motor impulsivity, that is the inability to inhibit a response. All participants viewed practice trials which showed example stimuli and indicated what response, if any, would be required. The task had 650 trials and lasted around 22 minutes.

Secondly, we used the Delay Discounting Test (DDT) to measure temporal impulsivity. This task has been previously shown to differentiate between participants with and without ADHD [68], be sensitive to acute interventions [69] and has been used to examine the effects of the serotonergic drug, fluoxetine, in ADHD populations [27]. Critically, discounting has been previously shown to be impacted by serotonin depletion, albeit in rats, which indicates that it may be sensitive to the effects of a TRP loading paradigm [70]. The task requires participants to make choices between hypothetical rewards now or at a point in the future. This process is repeated once for each delay with descending and ascending reward amounts and for several different future time points e.g., 1 week, 2 weeks, 1 month, 3 months, 6 months and 1 year. The task contained 364 trials and took around 12 minutes to complete. Area under the curve (AUC) was computed to provide a measure of impulsivity that is purely derived from the data and not based on assumptions about the form of the discount function [71, 72]. A higher value for AUC indicates reduced impulsivity.

The final task completed was the Iowa Gambling Task, which provides a measure of cognitive impulsivity and in which those with ADHD typically exhibit riskier choices as would be expected given the higher impulsivity in this group [73–75]. The type of decision making assessed in this task has previously been linked to serotonin synthesis, albeit in a population with borderline personality disorder [76]. Participants completed 100 trials and for each trial they chose a card from four choices (A-D); their choice of card determined whether they won or lost virtual money. Two decks gave large gains and large losses, resulting in an overall loss, and are considered riskier choices, whilst the other two decks give small gains and small losses, and resulted in an overall gain. High impulsivity is associated with more riskier deck choices. This can be measured over all 100 trials, but the early trials in the task (~40 trials) are typically associated with a period of learning [77]. As such, we used data from the final 40

trials to provide a percentage of risk decisions score and a net score, calculated by subtracting risky choices from safe choices, such that a more negative net scores indicates more risky decisions. The whole task takes approximately 15 minutes to administer. Given that participants could recall which decks were associated with specific gains and losses between the baseline and post-TRP time points, we counterbalanced the use of two different versions of the task where the risky versus non-risky decks were not the same.

## Statistical analysis

All analyses were completed using SPSS (IBM SPSS Statistics for Windows, Version 28.0) and performed with a two-tailed significance value of 5% and data was checked for normality using histograms, measures of skewness and kurtosis, and the Kolmogorov-Smirnov normality test. Data for omission and commission errors and hit reaction times from the TOVA were log transformed before statistical analysis. Descriptive data in the form of the mean ($M$) ± the standard deviation ($SD$) or count/percentage ($N$/%) were used to characterise the sample in terms of demographic and clinical variables with group comparisons made with ANOVAs and chi-square analyses. For treatment outcomes, analyses were conducted using a 3 (TRP: BAL, LOW, HIGH) x 3 (Group: Control, ADHD-M, ADHD-UM) by 2 (Time: pre, post) ANCOVA with age and ethnicity as covariates because these were found to differ between the three groups (see Study Population, Table 1).

## Results

### Study population

In total, 914 potential participants were screened, of which 457 were excluded for not meeting the eligibility criteria. Across all three groups, exclusions were noted where participants were following a restrictive diet or reported other health conditions. Within the Control group specifically the most common reason for exclusion was high ASRS scores, as has been found in

**Table 1. Baseline demographic and clinical characteristics of the two groups.**

| Characteristic | Control ($N$ = 59) | ADHD-M ($N$ = 48) | ADHD-UM ($N$ = 46) | $F$ or $X^2$ | Significance |
|---|---|---|---|---|---|
| Female gender N (%) | 28 (56.0) | 31 (64.6) | 35 (76.1) | 4.280 | 0.118 |
| Right-handed N (%) | 46 (92.0) | 44 (91.7) | 37 (80.4) | 3.911 | 0.141 |
| Ethnicity N (%) | | | | 16.572 | <0.001 |
| White | 22 (44.0) | 38 (79.6) | 35 (79.2) | | |
| Asian | 16 (32.0) | 2 (4.2) | 4 (8.7) | | |
| Black | 1 (2.0) | 1 (2.1) | 1 (2.2) | | |
| Mixed | 8 (16.0) | 5 (10.2) | 2 (4.3) | | |
| Other | 3 (6.0) | 2 (4.2) | 4 (8.7) | | |
| Age (years) M (SD) | 23.24 (3.93) | 25.98 (5.02) | 25.85 (4.70) | 5.625 | 0.004[a] |
| Education (years) M (SD) | 5.11 (2.74) | 5.28 (2.65) | 4.60 (2.23) | 0.907 | 0.406 |
| ASRS M (SD) | | | | | |
| ASRS Total | 20.34 (9.58) | 54.90 (8.16) | 53.36 (7.66) | 321.752 | <0.001[b] |
| ASRS-IA | 11.36 (4.72) | 29.31 (3.67) | 29.28 (3.58) | 130.421 | <0.001[b] |
| ASRS-HI | 8.98 (5.58) | 25.58 (5.35) | 24.17 (5.49) | 257.156 | <0.001[b] |

* Chi-square compared only white vs non-white due to small categories.

[a]Control were significantly younger than both ADHD groups, who did not differ from each other.

[b]Control had lower ASRS scores than both ADHD groups, who did not differ from each other.

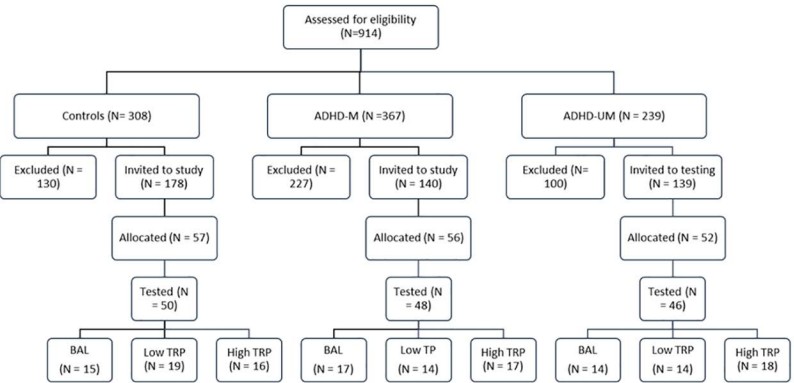

**Fig 1. Flow diagram showing screening and allocation of the different groups.** ADHD-M indicates medicated whilst ADHD-U indicates medication free.

other studies of this kind [69]. Within the ADHD groups, relatively few exclusions were made for low ASRS scores, with most exclusions being due medication use not fulfilling inclusion criteria, or the presence of comorbid conditions. Details of all exclusions are within S1 Table. All those eligible were invited to attend and around 30% responded and were allocated to a TRP condition. Between protein powder dispatch and testing, a small proportion passively withdrew from the study (i.e., did not confirm or attend appointment). This proportion was similar across groups (Control = 12.2%, ADHD-M = 14.3%, ADHD-UM = 11.5%). Details of the screening, allocation and group sizes are shown in Fig 1.

Demographic and clinical characteristics of the three groups are detailed in Table 1. As indicated, the three groups did not significantly differ for gender, handedness, or years in education. However, those in the ADHD groups were significantly older than the Control group, albeit by less than 3 years. Given the developmental trajectories of cognitive functions relevant to ADHD do not differ markedly in this period [78] this age difference is unlikely to be clinically meaningful. Additionally, ethnicity differed between the three groups with a higher proportion of white participants in the ADHD groups. As expected, and reinforced by our policy of exclusion based on ASRS score, the Control group had significantly lower ASRS scores than both ADHD groups, who did not differ from each other.

Within the ADHD-M group, 54.2% ($N = 26$) were receiving amphetamine, followed by 41.7% ($N = 20$) receiving methylphenidate. Just 4.2% ($N = 2$) reported use of non-stimulant medication, and in one of these cases, atomoxetine was combined with methylphenidate. The mean adherence to medication was 90.6% ($SD = 11.2$).

## Treatment outcomes

All participants completed all tasks at baseline and after consuming TRP. However, several task specific exclusions were made prior to analysis. Seven individuals were excluded from analysis in the DDT (Control = 1, ADHD-M = 2 ADHD-UM = 4) due to having >2 switch points at any delay at baseline [68]. Additionally, one individual (ADHD-UM) was excluded from the TOVA task due to a computer error during the baseline testing. No data was excluded for the IGT. Descriptive data for all outcome variables is shown in Table 2.

**Inattention.** After controlling for the covariates of ethnicity and age, analysis of omission errors showed no significant effect of Time ($F(1, 132) = 3.44$, $p = .066$, $\eta p^2 = .025$), TRP ($F(2, 132) = .2.02$, $p = .137$, $\eta p^2 = .030$) or Group ($F(2, 132) = 3.75$, $p = .055$, $\eta p^2 = .028$), although

**Table 2. Descriptive statistics for controls and those with ADHD, either taking medication (ADHD-M) or medication free (ADHD-UM), before and after differing levels of TRP loading.** All scores expressed as M(SD).

| | Attention | | | Motor Impulsivity | Cognitive Impulsivity | | Temporal Impulsivity |
|---|---|---|---|---|---|---|---|
| | Omissions | Hit RT | d' Prime | Commissions | Net Score | Risky Decisions | AUC |
| BAL | | | | | | | |
| Controls | | | | | | | |
| Pre | 3.79 (11.68) | 436.26 (40.18) | -0.56 (1.50) | 4.25 (3.80) | 1.47 (25.60) | 48.17 (31.99) | 0.42 (0.24) |
| Post | 4.72 (13.09) | 420.99 (71.10) | -0.31 (1.36) | 3.54 (2.39) | 3.20 (27.78) | 46.03 (34.78) | 0.45 (0.27) |
| ADHD-M | | | | | | | |
| Pre | 2.94 (3.33) | 489.61 (151.61) | 0.22 (1.35) | 5.52 (7.17) | -1.65 (20.39) | 52.15 (25.42) | 0.49 (0.36) |
| Post | 6.82 (12.18) | 455.21 (88.85) | -0.03 (1.35) | 3.99 (5.58) | -4.94 (20.48) | 56.18 (25.60) | 0.52 (0.33) |
| ADHD-UM | | | | | | | |
| Pre | 4.59 (5.05) | 492. 87 (81.40) | 0.07 (1.53) | 6.83 (10.31) | 9.85 (22.54) | 36.96 (28.91) | 0.45 (0.25) |
| Post | 6.30 (9.01) | 499.43 (158.70) | 0.19 (1.69) | 6.55 (10.63) | 6.42 (29.34) | 41.96 (36.68) | 0.50 (0.31) |
| Low TRP | | | | | | | |
| Controls | | | | | | | |
| Pre | 1.17 (2.29) | 438.78 (66.06) | -0.97 (1.39) | 3.52 (3.42) | -1.16 (20.31) | 51.53 (25.38) | 0.48 (0.29) |
| Post | 3.00 (6.15) | 443,29 (70.94) | -0.46 (1.12) | 4.80 (4.88) | 0.95 (29.33) | 50.39 (36.24) | 0.54 (0.30) |
| ADHD-M | | | | | | | |
| Pre | 5.25 (9.22) | 468.05 (110.13) | -0.25 (0.99) | 7.16 (8.61) | 6.14 (20.91) | 42.36 (26.16) | 0.47 (0.26) |
| Post | 11.90 (22.48) | 473.88 (116.43) | -0.23 (1.35) | 5.68 (4.68) | -5.29 (25.39) | 56.61 (31.74) | 0.50 (2.62) |
| ADHD-UM | | | | | | | |
| Pre | 3.30 (6.62) | 444.70 (66.09) | -0.30 (1.14) | 3.62 (3.03) | -5.86 (20.20) | 57.36 (25.25) | 0.37 (0.27) |
| Post | 1.69 (2.96) | 444.82 (86.28) | -0.69 (1.11) | 2.65 (1.77) | -11.14 (26. 90) | 63.93 (33.62) | 0.38 (0.25) |
| High TRP | | | | | | | |
| Controls | | | | | | | |
| Pre | 3.13 (5.31) | 456.09 (86.35) | -0.34 (1.68) | 2.78 (2.52) | 10.00 (25.19) | 38.47 (31.22) | 0.46 (0.27) |
| Post | 12.05 (22.05) | 465.12 96.96) | 0.24 (0.84) | 3.40 (3.49) | -1.25 (28.14) | 51.59 (35.14) | 0.46 (0.29) |
| ADHD-M | | | | | | | |
| Pre | 4.28 (7.51) | 485.09 (90.12) | -0.20 (1.30) | 5.92 (9.18) | -6.24 (19.39) | 57.82 (24.24) | 0.44 (0.24) |
| Post | 3.92 (6.68) | 478.83 (101.78) | -0.35 (1.40) | 2.87 (2.37) | 7.88 (18.43) | 40.15 (23.03) | 0.54 (0.26) |
| ADHD-UM | | | | | | | |
| Pre | 4.33 (4.29) | 479.83 (94.16) | -0.10 (1.14) | 7.16 (14.49) | 9.44 (16.62) | 38.19 (20.77) | 0.44 (0.30) |
| Post | 9.32 10.70) | 507.17 (134.92) | 0.47 (0.91) | 4.95 (5.03) | 0.78 (25.30) | 48.42 (31.61) | 0.45 (0.33) |

arguably the latter could be considered a trend towards significance with the Control group producing fewer errors than both ADHD groups ($p < 0.05$) as demonstrated by contrast analysis. There were no significant two- or three-way interactions ($p \geq .157$). For Hit Reaction Time, there was no significant main effect of Time ($F(1, 132) = .10$, $p = .758$, $\eta p^2 = .001$), TRP ($F(2, 132) = .1.07$, $p = .347$, $\eta p^2 = .016$) or Group ($F(2, 132) = 1,77$, $p = .175$, $\eta p^2 = .026$). There were no significant two- or three-way interactions ($p \geq .257$). For d' prime there were no significant main effects (Time $F(1, 132) = 1.21$, $p = .273$, $\eta p^2 = .009$), TRP ($F(2, 133) = 2.80$, $p = .064$, $\eta p^2 = .041$), Group ($F(1, 132) = 1.48$, $p = .232$, $\eta p^2 = .022$) or interactions ($p \geq .281$).

**Impulsivity.** For temporal impulsivity, after controlling for covariates, there was no significant main effect of Time ($F(1, 126 = 1.23$, $p = .270$, $\eta p^2 = .010$), TRP ($F(2, 126) = .02$, $p = .982$, $\eta p^2 = .000$) or Group ($F(2, 126) = .68$, $p = .510$, $\eta p^2 = .011$) on the AUC. There were also no significant two- or three-way interactions ($p \geq .187$). Similarly for cognitive impulsivity, for the percentage of risky decisions in the last 40 trials, there was no significant main effect of Time ($F(1, 133) = .52$, $p = .473$, $\eta p^2 = .004$), TRP ($F(2, 133) = 1.92$, $p = .150$, $\eta p^2 = .028$) or

Group ($F(2, 133) = .22$, $p = .803$, $\eta p^2 = .003$). There were no significant two or three-way interactions ($p \geq .197$). The Net Score for the last 40 trials revealed similar null effects (Time $F(1, 133) = .484$, $p = .488$, $\eta p^2 = .004$), TRP ($F(2, 133) = 1.86$, $p = .169$, $\eta p^2 = .027$), Group ($F(1, 133) = .20$, $p = .821$, $\eta p^2 = .003$), interactions $p \geq .150$). Finally, for motor impulsivity, measured by commission errors, there was no significant main effect of Time ($F(1, 132 = 1.19$, $p = .277$, $\eta p^2 = .009$), TRP ($F(2, 132) = .88$, $p = .419$, $\eta p^2 = .013$) or Group ($F(2, 132) = .96$, $p = .386$, $\eta p^2 = .014$). There were no significant two- or three-way interactions ($p \geq .157$).

## Discussion

The aim of this study was to establish the effects of acute TRP modulation, using a loading paradigm, on ADHD symptomology in adults with the condition. We hypothesized that TRP would have dose-dependent reduction effects on inattention and impulsivity. A control group of individuals without ADHD was included to provide a reference group The data demonstrated there were no significant effects of TRP on measures of inattention and impulsivity in any of the groups. The majority of prior TRP modulation studies focus on aggressive behaviours in ADHD [43] and there is a dearth of literature examining the impact of TRP modulation on the core symptoms of the condition. There are no studies examining impulsivity, as opposed to impulsive aggression, before and after TRP modulation, meaning the results of the current study, whilst indicating no effect of the TRP intervention, are novel. One prior study did consider behavioural inhibition in children with ADHD but focused on the impact on high and low-hostile groups following TRP depletion [49]. This study reported an increase in behavioural inhibition, implying improved impulsivity, following TRP depletion but this was only found for those with high levels of hostility, with opposing effects found for low hostile individuals [49]. In the current study, we did not measure trait aggression or hostility and therefore we cannot rule out the possibility that our lack of effects is due to high and low hostile groups combining and cancelling out differing effects. However, the lack of bimodal distribution in the data would suggest this is not the case.

One previous study did examine attentional processes using a CPT task, similar to the TOVA employed in the current study, in adults with ADHD using acute TRP depletion [48]. They reported an increase in omission errors under TRP depletion in both groups, thus indicating worsened attention after TRP depletion. Based on this result, we might have expected to see a reduction in omission errors in our ADHD groups, i.e., improved attention following TRP loading, which was not the case. However, the previous study also found shorter reaction times with TRP depletion, suggesting a speed-accuracy trade off and no overall change in attention, and therefore, more in line with our results. Another study examined lapses of attention in children with ADHD, and found these were greater under the balanced condition rather than depletion [47], implying better attention after TRP depletion, therefore contradicting the previous omission error findings [48]. In these previous studies, TRP depletion was used, raising the possibility of the results being impacted by a floor effect, given previous research has indicated lower levels of TRP in those with ADHD [34]. The current study utilised a loading paradigm, therefore preventing floor effects, which could explain the differences in the reported findings.

The lack of effects of TRP should be considered in the wider context of the sample. It is noteworthy that we did not find any significant baseline differences between the three groups which we would have expected given that the tasks were selected, in part, due to their prior ability to differentiate between those with and without ADHD [61–63, 68, 73, 74]. Considering first the TOVA task, previous work has found increased commission errors and slower reaction times [61, 63], as well as greater reaction time variability [62] in those

with ADHD. Omission errors have been found to be increased but not always significantly [61, 63]. Despite the lack of significant group differences here, our results do demonstrate similar patterns. For commission errors, baseline errors are higher in both groups with ADHD. Similarly, the reaction times are also slower in the ADHD groups in the present study. Our findings regarding omission errors are more complex. We found a lower level for medicated individuals relative to controls but a higher level for unmedicated individuals. Interestingly, the prior study which found no overall significant effect [61] combined medicated and unmedicated individuals whilst the one finding a significant increase in those with ADHD asked participants to abstain from their usual medication for testing. Therefore, for the TOVA task, whilst our results did not indicate significant differences between control participants and those with ADHD, the results are not fully at odds with previous work. For the DDT, previous research showed a significantly smaller AUC in those with ADHD [68]. However, the sample was small, with just 21 individuals declaring a diagnosis of ADHD, in contrast to 218 control individuals. Information pertaining to the medication status in this previous study was also lacking. Furthermore, a recent meta-analysis has demonstrated that whilst delay discounting is impacted in ADHD, the effect size is small and the studies included had very varied results [79]. It is possible that the differences in sample size and medication can explain the lack of group differences in the present study. Finally, for the IGT, previous work has reported a decrease in the net score in those with ADHD [73] and increased risky choices [74]. Our results paint a more nuanced picture with different directions of effect depending on medication status and all were non-significant. In the previous studies medication is either not mentioned [74] or participants were asked to abstain from their usual drug treatment prior to taking part [73], which may suggest that differences in the sample medication status underlie the lack of group differences we found here. Given the lack of differences between those with and without ADHD in the current study, we cannot rule out the possibility that the selected tasks were not appropriate. However, as the hypothesis relates to the impact of TRP loading rather than group differences per se, the absence of group differences is not critical to the aim of the study.

It is possible that the lack of modulation by TRP in any group is caused by the presence of ceiling or floor effects in the data. Comparison to previous studies suggests that this is unlikely to be the case. For example, previous TOVA task reporting indicates similar omission, commission and reaction time data [61]. For the DDT, similar results are found for control group elsewhere [68], although the ADHD group did have a lower AUC in the current study compared to previous work. This is unlikely to be due to education level as both studies included university level education and it should be noted that the variation in AUC for the ADHD group in the previous study was substantial. For the IGT task, previous data does suggest that all groups in our study took more risks than in some previous work [80], however, the baseline range of 37–58% risky decisions in the current study indicates plenty of scope for modulation and therefore a lack of ceiling or floor effects.

It is noteworthy that there were no significant differences in ASRS scores between the medicated and unmedicated individuals with ADHD. Based on previous research we anticipated that adults receiving medication would have a lower score than those who are unmedicated, especially given the adherence to medication reported [81]. However, it is also possible that only adults with very severe ADHD symptoms are choosing to take medication meaning their symptoms are still comparable to those not taking it. It is not possible to confirm this in the current study given the single time point at which the ASRS was collected (i.e., rather than before and after starting on medication). In any event, comparable ASRS levels between medicated and unmedicated groups have been found previously [69, 82], and medication has been found not to dictate long term outcomes using this scale [83] although there is also evidence

that the scale can differentiate between those who are medicated in a population from Norway [84].

The lack of impact of TRP on measures of attention and impulsivity is not entirely at odds with previous work examining the effects of TRP on measures other than aggression, with depletion studies failing to find any effects on mood [85], affective prosody [86] and verbal declarative working memory [87] which could suggest that any benefits of TRP modulation are limited to aggressive behaviours. Prior work with acute fluoxetine has also failed to find effects on AUC, reaction time or omission errors in those with ADHD utilising similar tasks to those employed in the current study [27], supporting this suggestion that treatments thought to impact on 5-HT, either through increased synthesis, and, therefore, release, or decreased reuptake, do not impact core symptoms as much as additional areas of impairment. Only one previous study investigating the effectiveness of an SSRI in managing ADHD reported improvements in core symptoms [24]. However, participants also had co-morbid depression, and as such it is unclear if these improvements are ADHD-specific. Additionally, whilst there is some evidence to indicate that the tasks, we selected are dependent on serotonin synthesis [48, 66, 70, 76], and therefore would be sensitive to TRP loading, it is possible that the altered synthesis was insufficient or did not result in altered release as would be expected [33] which may result in no effects of TRP loading. Whilst research indicates that oral administration of 5-HT pre-cursors can result in increased levels of central 5-HT, it is not clear if this results in a meaningful change in synaptic 5-HT, or whether release is re-calibrated by autoreceptors [88] which may render the increased release irrelevant. Furthermore, there is also suggestion that modulation of TRP may exert effects through non-5-HT routes, including via BDNF and NMDA [89], suggesting a greater understanding of TRP modulation is needed.

A suboptimal treatment regime could offer an alternative explanation for the lack of effects reported in the present study and previous studies with other serotonergic agents. Specifically, the TRP loading paradigm used in the present study may have been insufficient to impact the measures. Although the loading paradigm has previously been shown to impact on cognition [54], the previous study conducted tests of cognition after 3 hours, rather than the one hour period in the present study. It is therefore possible that the lack of effects were due to the testing happening too soon after administration. This idea is partially supported by other work, which failed to find cognitive effects of TRP loading, albeit using a different paradigm, after just one hour [55]. One possible explanation for this is that whilst the blood plasma levels of TRP would likely have peaked by one hour, based on previous work [54, 55], this may not have equated to sufficient rise in brain TRP levels. This seems unlikely given the close correlation of temporal changes in blood and brain TRP levels noted previously, however, given that all previous work was conducted in animals, it is possible this correlation is not present or as strong in humans [57–59]. It is also possible that the lack of effect of the paradigm is due to the sample. Previous work has included participants with multiple sclerosis [54] or healthy individuals [55] and this study is, to our knowledge, the first use of the paradigm in those with ADHD. It is noteworthy that the whey protein in which the TRP was consumed may have impacted transport of the protein to the brain and consequently the impact on 5-HT could have been small [45], even though plasma TRP was high. Whilst this would have been the case in the previous study using the same paradigm [54], the possibility of lower levels of 5-HT in those with ADHD may have meant that this loading paradigm did not raise brain 5-HT levels sufficiently to impact the measures. It is also possible that longer-term intervention is needed for loading to take effect on the outcomes measured here. In the current study, we employed an acute intervention, mirroring the frequently used depletion approaches. However, research has suggested that long-term TRP administration can enhance cognitive performance in rats [90] and as such a longer duration study may be appropriate. It has also been suggested that

inadequate control of autoregulatory processes that control 5-HT release may mean serotonergic treatments are not effective in ADHD [91]. Specifically, the 5-HT$_{1A}$ autoreceptors which typically reduce the firing rate of 5-HT neurons, and thus release of 5-HT may need to be antagonised alongside any treatment aimed to increase available synaptic 5-HT, for treatments to be effective. It is also possible that no differences were seen in the current study because there was no deficiency in TRP, or 5-HT in the sample. Previous work has linked changes in the TRP2 genes to ADHD [37, 38] but in large UK samples and European samples this link was not found [92, 93]. Additionally, dietary data indicates that those with ADHD can have normal levels of dietary TRP, meaning that there may not be a deficiency that can be reversed with TRP loading [35]. Future studies should consider additional genotyping and collecting dietary information.

The current study is, to our knowledge, the first study to employ a robust double-blind randomised controlled trial design to examine the core symptoms of ADHD following TRP modulation. No reports of breaking blinding were made by researchers. In addition to the design of the study, which is considered a gold standard for intervention testing [94], there are several other strengths to the work. Firstly, the study included a balanced sample of males and females, addressing the gender bias in previous studies and meeting recent calls for more research to include females with the condition [95]. Secondly, the study utilised an existing loading paradigm which has been verified with plasma measurements of TRP previously and shown to impact cognition [54]. Thirdly, we used existing validated measures of the core symptoms of ADHD. We also included individuals with ADHD taking medication and those not taking any to allow us to dissect any interactions with medication. Fourthly, the trial was conducted remotely over a video conferencing platform. Although remote trials pose their own challenges, such as digital skills and the engagement of individuals from less advantaged groups [96], they also come with a unique set of advantages. For example, online trials may increase the external validity of the trial as they more closely reflect the real-life circumstances in which participants would engage with the intervention if it were effective [97] and facilitate the recruitment of larger samples, especially from smaller groups [98]. Furthermore, a remote synchronous trial as the one presented here may strike the right balance between flexibility and convenience, a perceived benefit identified in previous research investigating participant experiences of internet trials, and lack of connectedness and understanding, a perceived disadvantage, since participants were assisted by a researcher throughout [99]. Lastly, despite that, the trial used controlled paradigms to measure attention and impulsivity, conducting the trial remotely in participants' own surroundings increases the ecological validity of the study and findings are more representative of how participants would perform in real-life conditions, where distractions and noise are ordinarily present [100].

Despite these strengths, there are some limitations to the work. Firstly, although we met the sample size required according to our initial power calculations, and this is the largest study to date examining TRP modulation in ADHD, the calculations assumed a medium effect and, as such, smaller effects may not have been detected with this size sample. The results reported here showed small effect sizes, which would require a considerably larger sample size (N = 387 with $f = 0.10$ to achieve the minimally acceptable power, 0.80). Additionally, although our sample size should have allowed detection of within-subjects effects, which required N = 63, as well as the interaction effects we powered for to test the hypothesis, according to power calculations for a medium effect, we did fall short of the required power for between-subject effects, which would have required a considerably larger sample N = 288. Secondly, due to the COVID-19 pandemic, all data was collected remotely. Although the participant and researcher were always visible to each other and the same platform was used to administer the tests as would be done in a laboratory, it is possible that the

non-standard environment in which the data was collected could have impacted the findings. Nonetheless, previous work comparing data quality of a laboratory and an online unassisted study found few differences between the two modalities in terms of attention [101] and another study comparing a laboratory and an online assisted study conducted over a video conferencing platform found no differences in performance [102]. Whilst this could be considered a benefit of the study because it moves away from laboratory to naturalistic settings as outlined above, it is also a limitation. Thirdly, and relating to the remote administration, we did not measure TRP plasma levels in the current study. Whilst a previous studying using this loading paradigm has measured TRP levels [54] it would have been ideal to collect this information, not least because the previous work was not in individuals with ADHD. Additionally, plasma levels of amino acids would have allowed us to check overall protein consumption. This would have allowed us to confirm whether participants had abstained from a protein rich meal the night before and on the test day. Given that testing took place throughout the day, and it is possible participants protein intake varied this additional information would have been beneficial, especially because there is an association between circadian rhythms, known to be altered in ADHD [103, 104] and TRP [105]. Furthermore, if TRP levels had been measured at baseline, the study could also have established whether those with ADHD had lower levels of TRP initially as has been found previously [34–36]. Fourthly, we did not measure hyperactivity in the present study, focusing on just attention and impulsivity and within a relatively narrow age group. It is therefore possible that the findings would not extend to hyperactivity and adults aged older than 35 years. Fifthly, we did not confirm ADHD diagnosis as part of the study. Instead we recruited those with an existing diagnosis and used the ASRS to confirm this [51]. The lack of a structured clinical interview for diagnosis is therefore a limitation of the study. Similarly, we did not confirm a lack of diagnosed conditions in the Control participants but asked them to self-report they had not received any diagnosis of ADHD or other excluded condition and complete the ASRS. It is therefore possible some individuals had undiagnosed conditions or gave false responses. Also related to diagnosis, we did exclude individuals with co-morbid conditions which reduces the generalisability of findings, given high rates of co-morbidity in ADHD [106]. Finally, this study only considered the acute effects of TRP loading and a longer-term study may be beneficial to determine the effects of long-term treatment. This would also allow a greater diversity of measures to be used. The discussion presented here demonstrates that the measures we used, whilst widely considered appropriate measures of the key constructs important in ADHD, can produce quite inconsistent results. A longer-term intervention would allow use of self-report symptom measures pre and post treatment and assessment of wider constructs such as emotional dysregulation and quality of life.

In conclusion, to our knowledge, this is the first study to adopt a remote double blind randomized controlled trial to examine the effects of TRP modulation on attention and impulsivity in adults with ADHD. The results indicate that there is no benefit to dietary supplementation with TRP for these core symptoms of the condition, although based on previous research, there may be benefits for aggressive behaviours. Future research should consider longer-term supplementation along with measures of plasma TRP levels, as well as a larger sample size and inclusion of individuals with co-morbid conditions.

## Supporting information

**S1 Table. Details of reasons for participant exclusion.** Note that many participants were excluded on multiple grounds. For ease we made exclusions based on the reasons listed in the order listed in the table below. Percentages reflect the percentage of the total number excluded for a group i.e., Control N = 130, ADHD Medicated (N = 227) and ADHD Unmedicated

N = 100, as per Fig 1. All percentages are given to the nearest whole number and therefore do not always sum to 100.
(DOCX)

**S1 Data.**
(PDF)

**S2 Data.**
(PDF)

**S3 Data.**
(PDF)

**S4 Data.**
(PDF)

## Acknowledgments

No funding was received to support this research, but we would like to acknowledge the assistance of dissertation students in collecting data throughout this study.

## Author contributions

**Conceptualization:** Larisa M. Dinu, Paul G. Overton, Eleanor J. Dommett.

**Data curation:** Larisa M. Dinu, Samriddhi N. Singh.

**Formal analysis:** Larisa M. Dinu, Alexandra L. Georgescu, Eleanor J. Dommett.

**Investigation:** Larisa M. Dinu, Samriddhi N. Singh, Neo S. Baker, Eleanor J. Dommett.

**Methodology:** Larisa M. Dinu, Alexandra L. Georgescu, Paul G. Overton, Eleanor J. Dommett.

**Project administration:** Larisa M. Dinu, Eleanor J. Dommett.

**Software:** Alexandra L. Georgescu.

**Supervision:** Larisa M. Dinu, Eleanor J. Dommett.

**Writing – original draft:** Eleanor J. Dommett.

**Writing – review & editing:** Larisa M. Dinu, Neo S. Baker, Alexandra L. Georgescu, Paul G. Overton, Eleanor J. Dommett.

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
