## [Decision Letter · Decision Letter 0]

2 Oct 2023

PONE-D-23-26114The effects of tryptophan loading on Attention Deficit Hyperactivity in adults: A remote double blind randomised controlled trialPLOS ONE

Dear Dr. Dommett,

Thank you for submitting your manuscript to PLOS ONE. After careful consideration, we feel that it has merit but does not fully meet PLOS ONE’s publication criteria as it currently stands. Therefore, we invite you to submit a revised version of the manuscript that addresses the points raised during the review process.

We look forward to receiving your revised manuscript.

Kind regards,

Mu-Hong Chen, M.D., Ph.D.

Academic Editor

PLOS ONE

Reviewers' comments:

Reviewer's Responses to Questions

**Comments to the Author**

1. Is the manuscript technically sound, and do the data support the conclusions?

Reviewer #1: Yes

Reviewer #2: Yes

Reviewer #3: Yes

2. Has the statistical analysis been performed appropriately and rigorously? 

Reviewer #1: Yes

Reviewer #2: Yes

Reviewer #3: Yes

3. Have the authors made all data underlying the findings in their manuscript fully available?

Reviewer #1: Yes

Reviewer #2: Yes

Reviewer #3: Yes

4. Is the manuscript presented in an intelligible fashion and written in standard English?

Reviewer #1: Yes

Reviewer #2: Yes

Reviewer #3: Yes

5. Review Comments to the Author

Reviewer #1: Review: PONE-D-23-26114

This is the first study to adopt a double blind randomized controlled trial to examine the effects of acute tryptophan (TRP) loading on attention and impulsivity in adults with ADHD. They found no effects of TRP were observed for any of the measures. The strength of this study is to use validated neuropsychological measures of the inattention and impulsivity, to include individuals with ADHD taking medication and those not taking any, and to utilize clear study protocol and TRP loading paradigm. Although the limitations of the current study are appropriately discussed, I still have some major concerns.

Comment 1

Although the authors stated that the study utilised an existing loading paradigm which has been verified with plasma measurements of TRP previously, the cognitive tasks were conducted 3 hours after dietary intake in a sample of patients with multiple sclerosis in the study mentioned by the author (1). Another study using acute tryptophan loading approach measured pharmacokinetics of the acute tryptophan intervention showing that TRP reached maximum plasma levels of free tryptophan at 3 hours, and connectivity changes of resting‐state MRI was observed 3 hours after the ingestion (2). Additionally, the effect of TRP loading could have been small to the brain, even though plasma TRP was high. However, the authors choose to measure inattention and impulsivity again 1 hour followed by consumption of a protein drink. Could the authors explain why they decide to wait only1 hour before completing the tests again? Is there more evidence supporting the study design?

Ref 1: Lieben CK, Blokland A, Deutz NE, Jansen W, Han G, Hupperts RM. Intake of tryptophan-enriched whey protein acutely enhances recall of positive loaded words in patients with multiple sclerosis. Clin Nutr. 2018 Feb;37(1):321-328.

Ref 2: Deza-Araujo YI, Neukam PT, Marxen M, Müller DK, Henle T, Smolka MN. Acute tryptophan loading decreases functional connectivity between the default mode network and emotion-related brain regions. Hum Brain Mapp. 2019 Apr 15;40(6):1844-1855.

Reviewer #2: The authors conducted a randomized double blind study to investigate manipulation of plasma tryptophan halt on measures of inattention and impulsivity in healthy participants and medicated and unmedicated ADHD patients between the age of 18-35. To this end, they used three paradigms to measure inattention and impulsivity related outcome variables.

The study is generally well conducted in its rigor, and well described in the method part of the manuscript, but a number of concerns and inconsistencies weaken the conclusions.

General comments

- What is the basic hypothesis of the study? This is not clearly stated

o Are the selected outcome variables, as challenged by the experimental paradigm, serotonin synthesis-dependent?

o Signal substances are prefabricated and stored in dense-core vesicles, with a large releasable pool of pre-synthesized transmitter. All serotonin dependent functions under all experimental conditions cannot simply be assumed to be synthesis dependent.

o The authors thus need to clearly state a hypothesis and discuss (or cite literature, if there is any) whether the outcome variables they chose to measure, as challenged by experimental paradigm they employ, are expected to be dependent on ongoing serotonin synthesis. This seems to be a tacit assumption, but it has to be clearly spelled out in the introduction

o For example, studies have shown that cognitive control and stress response under certain conditions are dopamine synthesis dependent.

o Authors thus need to cite literature establishing some sort of correlation / dependency between serotonin synthesis and the measured outcome variables, or at the very least discuss this in the introduction together with their hypothesis.

- The ASRS scores in medicated and unmedicated ADHD patients are almost exactly the same, which is rather strange with 90% treatment adherence as they report in the results section (line 306).

o CS medications have high response rate (70-80%), when response is defined as a minimum 30% reduction in symptom scores. With 90% adherence, you would expect to see a statistically significant difference between medicated and unmedicated ADHD patients, otherwise the authors could also conclude from their study that medication (amphetamine and methylphenidate) is ineffective in reducing ADHD symptom scores.

o Authors need to explain this discrepancy, which may put into question the validity of their study and indicate a flaw in the design.

- Authors correctly claim that some of the paradigms used in the study are robust enough to differentiate controls from ADHD patients, but this doesn’t seem to be the case in present study as the authors report no differences between the groups even in the pretreatment phase (baseline).

o If there were any group differences at baseline between healthy controls and ADHD patients, these should be reported as otherwise the paradigms might not have given expected results in the first place, i.e. before TRP modification is even employed

- If medication can not differentiate between treated and untreated patients and robust paradigms cannot differentiate between controls and cases, the value of the finding that TRP had no effect gets weakened in my opinion.

Minor details

- On line 59, authors mention “limited efficacy” of CS medication as a rationale for testing other potential interventions, yet on line 51 mention that 80% respond to CS medication. Authors should either explain how 80% response rate indicates “limited efficacy” or rephrase the statements.

- On line 55-56, authors state that “some patients experience only a 30% symptom reduction, demonstrating treatment needs are not always met”, but that is often the definition criterium for response (30% reduction). This should be rephrased to make it more consistent with the definition of response in the literature.

Reviewer #3: In this manuscript, the authors examine the effect of acute TRP modulation on ADHD symptoms in adults, focusing on attention and impulsivity. They performed a remote double-blind trial with 3 TRP loading doses (balanced, low, and high) in adults with and without ADHD symptoms. Their results showed no effects of TRP loading on the relevant outcome measures of attention and impulsivity in any of the groups.

This is an interesting and timely study, which addresses an important and novel question in the adult ADHD literature. The methodology is generally sound, and the results are clearly described. There are however a few limitations that somewhat reduce my enthusiasm from the manuscript, which should be addressed in a revision. Specifically:

1. Introduction and rationale for the study:

- The introduction nicely outlines the rationale for the potential involvement of 5-HT in ADHD and why the concentration of TRP can be relevant. However, from the introduction it is not clear that a one-time consumption of TRP may have any effect, and that it is more about a long-term diet. The authors should better explain why they hypothesize that a one-time consumption may have an effect on behavioral tests as the ones used here. In the 2017 paper cited for the interval used (ref [52]) it is mentioned that executive functions did not change in this interval – why did the authors hypothesize that they would?

- Also the design does not allow us to know whether participants with ADHD had lower TRP levels to begin with. If not – why would one assume that TRP loading would be beneficial?

2. Methods:

- The clinical trial was pre-registered, and no changes were made to the registered protocol, which is good. Still, there are some weaknesses in the methodology, mainly stemming from the remote application of the trial. For example, the authors needed to rely on self-report regarding protein consumption the night before and did not have any physiological measures to examine baseline or post-treatment differences. These are significant weaknesses in the methodology that should be acknowledged and reduce from the potential impact of the outcome.

- The description of what does the 3 X 3 X 2 design mean should come earlier in the text – only in the statistical analysis section it becomes clear, but should be made clearer before (e.g., on p.5).

- Power analysis – I am a bit unclear about how the power was derived. Which calculator was used for it? And for the effect sizes used – which comparisons were taken, given the multiple groups here?

- Inclusion criteria – the participants did not go through SCID to confirm current ADHD diagnosis – this is a limitation that should be noted. Also, excluding participants over 35 due to age-related cognitive decline seems a bit excessive – please reconsider phrasing.

- When ‘gender’ is mentioned – do the authors refer to ‘sex’?

- The outcome measures are standard and are appropriate to address the question. However, performance-based metrics are notorious for having inconsistent effects in ADHD, and performance may be affected by additional factors. An inclusion of an additional self-report measure of attention may have been helpful.

- Participants confirmed that they had not consumed protein-rich meals the night before; However, was intervention time fixed for all subjects? Adult ADHD is known to be associated with delayed circadian rhythmicity (see Becker, 2020; Snitselaar et al., 2017) and there is an association between circadian-related time of day and TRP (e.g., Rao et al., 1994). This is needed to be further clarified.

- Statistical analysis: given the null results, I wonder if a Bayesian analysis to confirm no difference would be relevant here.

3. Results: the results are generally well described. A few points to consider:

- The Study Population section describes the exclusion / drop out reasons only vaguely – please provide accurate numbers (and %) for the different exclusion reasons. This may be important for replication purposes etc.

- I wonder if the results of any of the tests have not changed due to floor/ceiling effects, due to the young adult sample. Have the authors compared their data to published norms of these standardized tests? In other words, if they were already at peak performance to begin with, not much change is expected, especially given task learning that may happen from one administration to the other.

- The study is a double-blind RCT, but it is not clear if the blinding procedure was successful: have the authors examined blinding success with participants and/or staff? This may not be critical due to the null effects, but still important to report in a double-blind RCT.

4. Discussion: limitations should mention the lack of more objective physiological measures and the fact that only a single consumption and immediate effects were examined, among others.

6. PLOS authors have the option to publish the peer review history of their article (what does this mean?). If published, this will include your full peer review and any attached files.

Reviewer #1: **Yes: **Huey-Ling Chiang

Reviewer #2: **Yes: **Mussie Msghina

Reviewer #3: No

---

## [Author Response · Author response to Decision Letter 0]

15 Oct 2023

Please see attached file 'Response to Reviewers'

---

## [Decision Letter · Decision Letter 1]

30 Oct 2023

PONE-D-23-26114R1The effects of tryptophan loading on Attention Deficit Hyperactivity in adults: A remote double blind randomised controlled trialPLOS ONE

Dear Dr. Dommett,

Thank you for submitting your manuscript to PLOS ONE. After careful consideration, we feel that it has merit but does not fully meet PLOS ONE’s publication criteria as it currently stands. Therefore, we invite you to submit a revised version of the manuscript that addresses the points raised during the review process.

We look forward to receiving your revised manuscript.

Kind regards,

Mu-Hong Chen, M.D., Ph.D.

Academic Editor

PLOS ONE

Journal Requirements:

Reviewers' comments:

Reviewer's Responses to Questions

**Comments to the Author**

1. If the authors have adequately addressed your comments raised in a previous round of review and you feel that this manuscript is now acceptable for publication, you may indicate that here to bypass the “Comments to the Author” section, enter your conflict of interest statement in the “Confidential to Editor” section, and submit your "Accept" recommendation.

Reviewer #1: All comments have been addressed

Reviewer #2: All comments have been addressed

Reviewer #3: All comments have been addressed

Reviewer #4: (No Response)

2. Is the manuscript technically sound, and do the data support the conclusions?

Reviewer #1: Yes

Reviewer #2: Yes

Reviewer #3: Yes

Reviewer #4: Yes

3. Has the statistical analysis been performed appropriately and rigorously? 

Reviewer #1: Yes

Reviewer #2: Yes

Reviewer #3: Yes

Reviewer #4: Yes

4. Have the authors made all data underlying the findings in their manuscript fully available?

Reviewer #1: Yes

Reviewer #2: (No Response)

Reviewer #3: Yes

Reviewer #4: Yes

5. Is the manuscript presented in an intelligible fashion and written in standard English?

Reviewer #1: Yes

Reviewer #2: Yes

Reviewer #3: Yes

Reviewer #4: Yes

6. Review Comments to the Author

Reviewer #1: The authors have made a sincere effort to be responsive to the previous review, and also made additional clarifications to the manuscript which have improved the clarity of the presentation.

Reviewer #2: (No Response)

Reviewer #3: The authors did a great job addressing all comments made by the reviewers. I have no further comments.

Reviewer #4: A 3-way factorial designed randomized controlled clinical trial was conducted which aimed to evaluate the effect of acute tryptophan (TRP) loading on attention and impulsivity in adults with ADHD. No effects of TRP were observed.

Minor revisions:

1- Line 292: Define M.

2- State and justify the study’s target sample size with a pre-study statistical power calculation. The power calculation should include: (1) the estimated outcomes in each group; (2) the α (type I) error level; (3) the statistical power (or the β (type II) error level; (4) the target sample size, (5) the statistical testing method, and (6) for continuous outcomes, the standard deviation of the measurements.

3- Cite the statistical software used for the analysis.

4- Age was statistically significantly different between the Controls and both ADHD groups. Comment if the statistical difference is clinically meaningful. Oftentimes when sample sizes are large, age is statistically significantly different but the difference is not clinically meaningful.

5- Line 331: The standard statistical term for average is mean.

6- Table 2 title: Summary data has been expressed “M (SD)” rather than “M +/- SD” to be consistent with the format used in the table.

7. PLOS authors have the option to publish the peer review history of their article (what does this mean?). If published, this will include your full peer review and any attached files.

Reviewer #1: No

Reviewer #2: **Yes: **Mussie Msghina

Reviewer #3: No

Reviewer #4: No

---

## [Decision Letter · Decision Letter 2]

13 Nov 2023

The effects of tryptophan loading on Attention Deficit Hyperactivity in adults: A remote double blind randomised controlled trial

PONE-D-23-26114R2

Dear Dr. Dommett,

We’re pleased to inform you that your manuscript has been judged scientifically suitable for publication and will be formally accepted for publication once it meets all outstanding technical requirements.

Kind regards,

Mu-Hong Chen, M.D., Ph.D.

Academic Editor

PLOS ONE

Additional Editor Comments (optional):

Reviewers' comments:

Reviewer's Responses to Questions

**Comments to the Author**

1. If the authors have adequately addressed your comments raised in a previous round of review and you feel that this manuscript is now acceptable for publication, you may indicate that here to bypass the “Comments to the Author” section, enter your conflict of interest statement in the “Confidential to Editor” section, and submit your "Accept" recommendation.

Reviewer #4: All comments have been addressed

2. Is the manuscript technically sound, and do the data support the conclusions?

Reviewer #4: (No Response)

3. Has the statistical analysis been performed appropriately and rigorously? 

Reviewer #4: (No Response)

4. Have the authors made all data underlying the findings in their manuscript fully available?

Reviewer #4: (No Response)

5. Is the manuscript presented in an intelligible fashion and written in standard English?

Reviewer #4: (No Response)

6. Review Comments to the Author

Reviewer #4: (No Response)

7. PLOS authors have the option to publish the peer review history of their article (what does this mean?). If published, this will include your full peer review and any attached files.

Reviewer #4: No

---

## [Editor Report · Acceptance letter]

22 Nov 2023

PONE-D-23-26114R2 

The effects of tryptophan loading on Attention Deficit Hyperactivity in adults: A remote double blind randomised controlled trial 

Dear Dr. Dommett:

I'm pleased to inform you that your manuscript has been deemed suitable for publication in PLOS ONE. Congratulations! Your manuscript is now with our production department. 

Kind regards, 

on behalf of

Dr. Mu-Hong Chen 

Academic Editor

PLOS ONE